# Aqueous pyruvate partly dissociates under deep ultraviolet irradiation but is resilient to near ultraviolet excitation

Jan Thøgersen [1], Fani Madzharova[1], Tobias Weidner [1] & Frank Jensen [1] ✉

The deep ultraviolet photochemistry of aqueous pyruvate is believed to have been essential to the origin of life, and near ultraviolet excitation of pyruvate in aqueous aerosols is assumed to contribute significantly to the photochemistry of the Earth's atmosphere. However, the primary photochemistry of aqueous pyruvate is unknown. Here we study the susceptibility of aqueous pyruvate to photodissociation by deep ultraviolet and near ultraviolet irradiation with femtosecond spectroscopy supported by density functional theory calculations. The primary photo-dynamics of the aqueous pyruvate show that upon deep-UV excitation at 200 nm, about one in five excited pyruvate anions have dissociated by decarboxylation 100 ps after the excitation, while the rest of the pyruvate anions return to the ground state. Upon near-UV photoexcitation at a wavelength of 340 nm, the dissociation yield of aqueous pyruvate 200 ps after the excitation is insignificant and no products are observed. The experimental results are explained by our calculations, which show that aqueous pyruvate anions excited at 200 nm have sufficient excess energy for decarboxylation, whereas excitation at 340 nm provides the aqueous pyruvate anions with insufficient energy to overcome the decarboxylation barrier.

Pyruvic acid and its conjugated base, pyruvate, is believed to have been among the first organic molecules in the prebiotic waters on Earth and essential to the origin of life [1–4]. The acid constant of pyruvic acid is $pK_a = 2.5$ and has been measured to be as low as $pK_a = 0.7$ at water–air interfaces [5,6]. Hence, in many aqueous environments and in particular on water surfaces and saline solutions [7] pyruvic acid is deprotonated and takes the form of pyruvate shown in Fig. 1a. Without the protection of oxygen and ozone, the molecules in the prebiotic waters were exposed to deep ultraviolet light limited only by water's absorption with an onset around $\lambda < 195\,nm$ [8,9]. The deep-UV irradiation photolysed many organic compounds like pyruvic acid and pyruvate thereby enabling the formation of new species in the prebiotic soup. The photochemistry of these species has so far been investigated using experiments without time resolution. Accordingly, the order of appearance of the detected products is unknown as they may result from repeated photo-excitations and secondary reactions with other photo-products [1,2,4]. Thus, the understanding of the photochemistry of

pyruvate and its products rests on an uncertain foundation. In the present work, we use time-resolved spectroscopy with sub-picosecond time resolution to identify the primary photoproducts and determine the dissociation quantum yield of aqueous pyruvate following deep-UV excitation. The experiments break uncharted territory, as no previous deep-UV experiments on aqueous pyruvate exist. The results we present form a solid basis for the prediction of the chemistry involving pyruvate in prebiotic waters.

The photochemistry of aqueous pyruvate continuous to be important to life on Earth [4,6,7,10–16], as it is estimated that plants and burning of biomass emit 0.85 Tg of pyruvic acid to the atmosphere every year, primarily in the aqueous phase [17]. Pyruvic acid is present as atmospheric organic aerosols in aqueous microdroplets and in large bodies of water like oceans and lakes, where it takes the form of pyruvate. Despite the fact that today's ozone layer is protecting the Earth from deep-UV irradiation, aqueous pyruvate is excited by sunlight reaching the Earth's atmosphere, because pyruvate has a weak near-UV

[1]Department of Chemistry, Aarhus University, Langelandsgade 140, DK-8000 Aarhus C, Denmark. ✉e-mail: frj@chem.au.dk

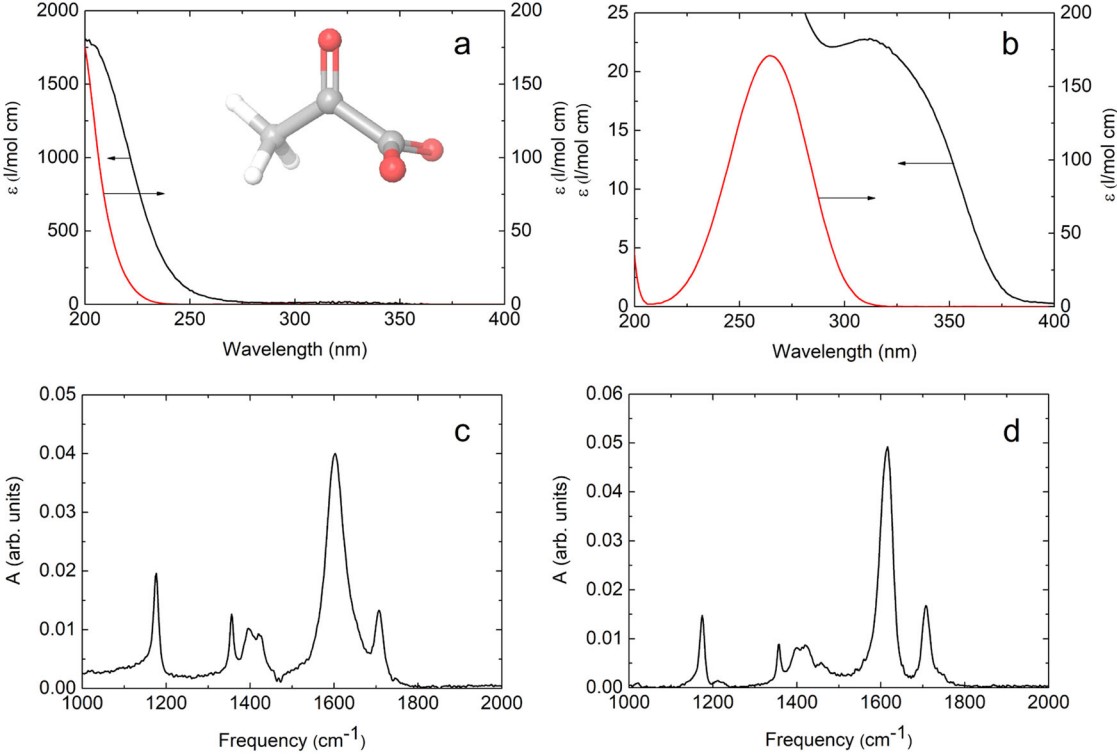

**Fig. 1 | Ultraviolet and infrared absorption spectra of aqueous pyruvate. a** UV absorption spectrum of aqueous pyruvate (black) compared to the absorption spectrum of aqueous propionate (red). Insert shows the ground state configuration of pyruvate. **b** UV absorption spectrum of aqueous pyruvate (black) compared to the absorption spectrum of aqueous acetone (red). IR steady state absorption spectra of pyruvate dissolved in **c** $H_2O$ and **d** $D_2O$ recorded against $H_2O$ and $D_2O$ references, respectively. Source data are provided as a Source Data file.

absorption band, in addition to its strong deep-UV absorption band. Considering the abundance of pyruvate, the near-UV photochemistry of pyruvate is potentially an important source of photoproducts. However, while the near-UV photolysis of neat and aqueous pyruvic acid has been studied extensively[1,2,4,18], studies of the photochemistry of aqueous pyruvate are scarce. Recently, photoelectron spectroscopy studies of gas-phase pyruvate have led Clarke et al.[10] to propose that the following chemical processes are induced by excitation at 290–380 nm:

$$h\nu_{UVA} + CH_3COCOO^- \rightarrow CH_3CO^- + CO_2 \qquad (1)$$

$$CH_3CO^- \rightarrow CH_3^- + CO \qquad (2)$$

$$CH_3^- \rightarrow CH_3 + e^- \qquad (3)$$

An even more recent study of the near-UV photolysis of pyruvate in small water clusters by Cao et al.[13], indicates that adding just a single water molecule to pyruvate significantly suppresses the dissociation channels, and addition of a second water molecule completely blocks all dissociation pathways. Hence, in contrast to efficient near-UV photolysis of pyruvic acid, the experiments with pyruvate in small water clusters suggest that aqueous pyruvate is not dissociated by near-UV light. These findings are consistent with a much lower photolysis rate of aqueous pyruvic acid observed when increasing the pH and also in line with the much lower photolysis yields observed in sodium pyruvate solutions relative to that of aqueous pyruvic acid[5,19].

The very different outcomes of pyruvate's gas-phase and cluster-phase photochemistry, naturally raises the question of pyruvate's photochemistry in bulk water: Are we to anticipate the pronounced disintegration observed in gas-phase pyruvate, or does the

photochemistry of pyruvate in bulk water resemble the near-UV photo-resiliency seen in small water clusters? Clearly, such information is needed to understand pyruvate's photochemistry in aerosols and thereby improve the predictive accuracy of climate models. In the present work we use time resolved spectroscopy with sub-picosecond time resolution to determine the primary quantum yield and photoproducts of aqueous pyruvate when excited in the deep- and near-UV bands. The results complete the foundations of pyruvate's primary near-UV photochemistry in gas-, cluster- and aqueous phase.

## Results and discussion

### Pyruvate's UV and IR absorption spectra

We begin by selecting the two wavelengths used for deep-UV and near-UV excitation of the aqueous pyruvate molecules in our experiments. Figure 1a, b shows the absorption spectrum of aqueous pyruvate. The spectrum is compared with the absorption spectra of aqueous acetone and aqueous propanoate. The comparison indicates that the UV absorption of pyruvate to a good approximation can be considered as a weak interaction between a localized carbonyl ($n_{C=O} \rightarrow \pi_{CO}^*$) chromophore with acetone as a model, and a localized carboxylate ($n_O \rightarrow \pi_{CO}^*$) chromophore with propanoate as a model. Calculations at the $\omega$B97X-D/aug-pcseg-1 level confirm this picture, with absorption maxima at 268 nm and 215 nm for acetone and propanoate, respectively, while the two lowest excitations for pyruvate are calculated at 298 and 215 nm. The calculated structure of ground state pyruvate has a 90° torsional angle between the carbonyl and carboxylate groups, as shown in the insert of Fig. 1a. The calculated rotational barrier for achieving a planar pyruvate configuration, however, is only 5 kJ/mol, and the planar structure has a strong interaction between the two chromophores, with calculated excitations at 352 and 282 nm. For structures with torsional angles between 0° and 90°, the excitations occur to a mixture of the anti-bonding $\pi_{CO}^*$ orbitals on the carbonyl

and carboxyl moieties, and the experimental spectrum represents a Boltzmann average over different torsional angles. The thermally averaged UV spectrum thus has a long wavelength absorption with strong carbonyl character and a shorter wavelength absorption with strong carboxylate character, as indeed observed in Fig. 1a, b. In the experiments investigating the photochemistry of aqueous pyruvate following deep-UV irradiation, we excite the pyruvate molecules at 200 nm as this wavelength results in the strongest UV absorption and the highest excitation energy, while still being able to pass through water without being absorbed to a significant level. In the experiments studying the near-UV photochemistry, we excite the pyruvate molecules at 340 nm, as this wavelength yields a strong overlap between pyruvate's absorption spectrum and the spectrum of the sunlight reaching the atmosphere. The photoreactions of pyruvate induced by the UV excitation are followed by time-resolved infrared spectroscopy. Figures 1c and 1d show the measured steady-state infrared absorption spectrum of aqueous pyruvate between 1000 $cm^{-1}$ and 2000 $cm^{-1}$ in $H_2O$ and $D_2O$. These spectra are used as a reference for the analysis of the time-resolved data. The most prominent features are the $C=O$ stretch transition at 1707 $cm^{-1}$ and the asymmetric $COO^-$ stretch transition at 1603 $cm^{-1}$. The absorption of the latter transition is wider in $H_2O$ than in $D_2O$ because of stronger hydrogen bonding to the solvent OH groups. The absorption at 1421 $cm^{-1}$ is assigned to the $C$-$COO^-$ stretch transition, while the absorption around 1397 $cm^{-1}$ pertains to a combination of symmetric $COO^-$ stretch and symmetric $CH_2$ bend transitions. The sharp absorption at 1356 $cm^{-1}$ is assigned to the excitation of the $CH_3$ umbrella vibration, while the absorption at 1176 $cm^{-1}$ is due to the $C$-$CH_3$ stretch transition.

## Transient infrared absorption dynamics of pyruvate after excitation at 200 nm

Figure 2 shows the transient absorption spectra $\Delta A(v,t)$ between 1030 $cm^{-1}$ and 2420 $cm^{-1}$ following excitation at 200 nm. Negative $\Delta A$ values imply the excitation of ground-state pyruvate molecules, while positive $\Delta A$ values indicate the formation of new species. All infrared transitions of ground state pyruvate appear in Fig. 2 as negative induced absorption peaks: The $C$-$CH_3$ stretch transition is at 1176 $cm^{-1}$, the $CH_3$ umbrella transition is observed at 1356 $cm^{-1}$, while the transition pertaining to a combination of symmetric $COO^-$ stretch and symmetric $CH_2$ bend transitions, expected at 1397 $cm^{-1}$, merges with the $C$-$COO^-$ stretch transition at 1421 $cm^{-1}$ to give the wide negative absorption with minimum at 1418 $cm^{-1}$. The asymmetric $COO^-$ stretch transition is observed at 1603 $cm^{-1}$ and the $C=O$ stretch transition at 1707 $cm^{-1}$. All the negative transients drop to a minimum immediately after the excitation pulse and then recover to about 20% of their initial value within 100 ps. The sub-picosecond drop in the transient absorption reflects the photoinduced transition of pyruvate anions from the electronic ground state to the excited electronic states, while the subsequent absorption recovery signifies the return of the excited pyruvate anions to an equilibrated population distribution of the vibration levels in the electronic ground state. Close inspection of the low-frequency side of the transition at 1176 $cm^{-1}$, the low-frequency side of the combined absorption with a minimum at 1418 $cm^{-1}$, and the low-frequency side of the transition at 1603 $cm^{-1}$ reveals the spectral dynamics characteristic of vibrational relaxation. The negative absorption associated with the asymmetric $COO^-$ stretch transition at 1603 $cm^{-1}$ prevents the observation of vibrational relaxation in the $C=O$ stretch mode at 1707 $cm^{-1}$. Due to the anharmonicity of the pyruvate ground state potential the vibrationally excited anions returning from the electronic excited states absorb at lower frequencies than the equilibrated ground state. As the pyruvate anions relax with time, the absorption shifts towards higher frequencies and merges with the ground state absorption at ~10 ps. The absorption of the vibrational relaxing anions is relatively weak, suggesting that the vibrational relaxation occurs on a shorter time scale than the return dynamics. If so, the measured transient absorption

dynamics reflect the concentration dynamics of the pyruvate ground state anions rather than spectral changes of the involved species, and we can thus use the transient absorption data to derive the photodissociation quantum yields. In order to strengthen this assessment, we have recorded the vibrational relaxation of the fundamental asymmetric carboxylate stretch and the carbonyl stretch by two-dimensional infrared (2D-IR) spectroscopy.

## 2D-IR spectra of pyruvate

Figure 3 shows 2D-IR pump-probe transmission spectra from 1540 $cm^{-1}$ to 1760 $cm^{-1}$ for pyruvate dissolved in $D_2O$. The absorption peaks in the linear absorption spectrum from Fig. 1d is reflected in the negative (blue) ground state population depletions on the dashed diagonal, while the positive (red) peaks show the induced absorption from the populations in the corresponding first excited vibration states. The off-diagonal peaks indicate that the excitation of one vibration mode affects the excitation of other vibration modes. The vibrational relaxation times of the first excited asymmetric carboxylate stretch state, and the first excited carbonyl stretch state are measured by varying the time delay between the pump and probe pulses. Figure 3 shows the 2D-IR spectrum of pyruvate for pump-probe delays of 0 fs, 500 fs, 1000 fs and 1500 fs. Figure 3a with a pump–probe delay of 0 fs only shows two diagonal features: the strong asymmetric carboxylate stretch transition at about 1603 $cm^{-1}$ and the weaker carbonyl stretch transition around 1707 $cm^{-1}$. Figure 3b–d shows that the intensity of the diagonal signals reflecting the population of the first excited states decreases with time. Figure 4 shows the integrated intensity of the absorption peaks in the diagonal of the 2D-IR spectra in Fig. 3 as a function of time. The data are well described by single exponential decays and show that the relaxation of the first excited state of the asymmetric carboxylate stretch occurs with a time constant of $0.6 \pm 0.1$ ps, while that of the carbonyl stretch is $1.2 \pm 0.1$ ps. The off-diagonal signals indicate a strong coupling between the carboxylate and carbonyl vibrations. They appear in 500 fs before they too decrease on time scales like those of the individual vibrations. As the vibrational relaxation rate and the strength of the vibrational coupling generally increases with the anharmonicity of the potential energy surface at higher excitation energies, the vibrational relaxation of pyruvate following the return to the electronic ground state is expected to occur in only a few picoseconds. This implies that the ~10 ps time scale for the ground state absorption recovery presented in Fig. 2 predominantly indicates the timescale for the return to the electronic ground state rather than the vibrational relaxation time of the pyruvate molecules returning to the ground state.

## Photodissociation quantum yield of pyruvate following excitation at 200 nm

The susceptibility of pyruvate to the 200 nm excitation may be quantified by its primary photodissociation quantum yield. Provided the pyruvate ground state is the only absorber at a particular frequency, the absorption dynamics at that frequency is proportional to the concentration dynamics of ground state pyruvate. Since the absorption of one photon excites one pyruvate anion, we can derive the primary photodissociation quantum yield of aqueous pyruvate 100 ps after the excitation as the induced change in the absorption after 100 ps relative to the initial absorption change. To a good approximation, the $C$-$CH_3$ stretch transition of aqueous pyruvate is the only absorber at 1176 $cm^{-1}$. From the transient absorption data at 1176 $cm^{-1}$ shown in Fig. 5a, we derive the primary quantum yield of $\Phi(t = 100$ ps$) = \Delta A(t = 100$ ps$)/\Delta A(t = 0.6$ ps$) = 19\% \pm 5\%$. The absorption data are well approximated by a single exponential function $\Delta A = -0.50$ mOD $\times \exp(-t/6.5$ ps$) -0.065$ mOD, indicating that the excited pyruvate anions return to the ground state on a $6.5 \pm 0.3$ ps time. The transient absorption data show that the dissociation yield remains constant for the rest of the 560 ps of the measurement. Consequently,

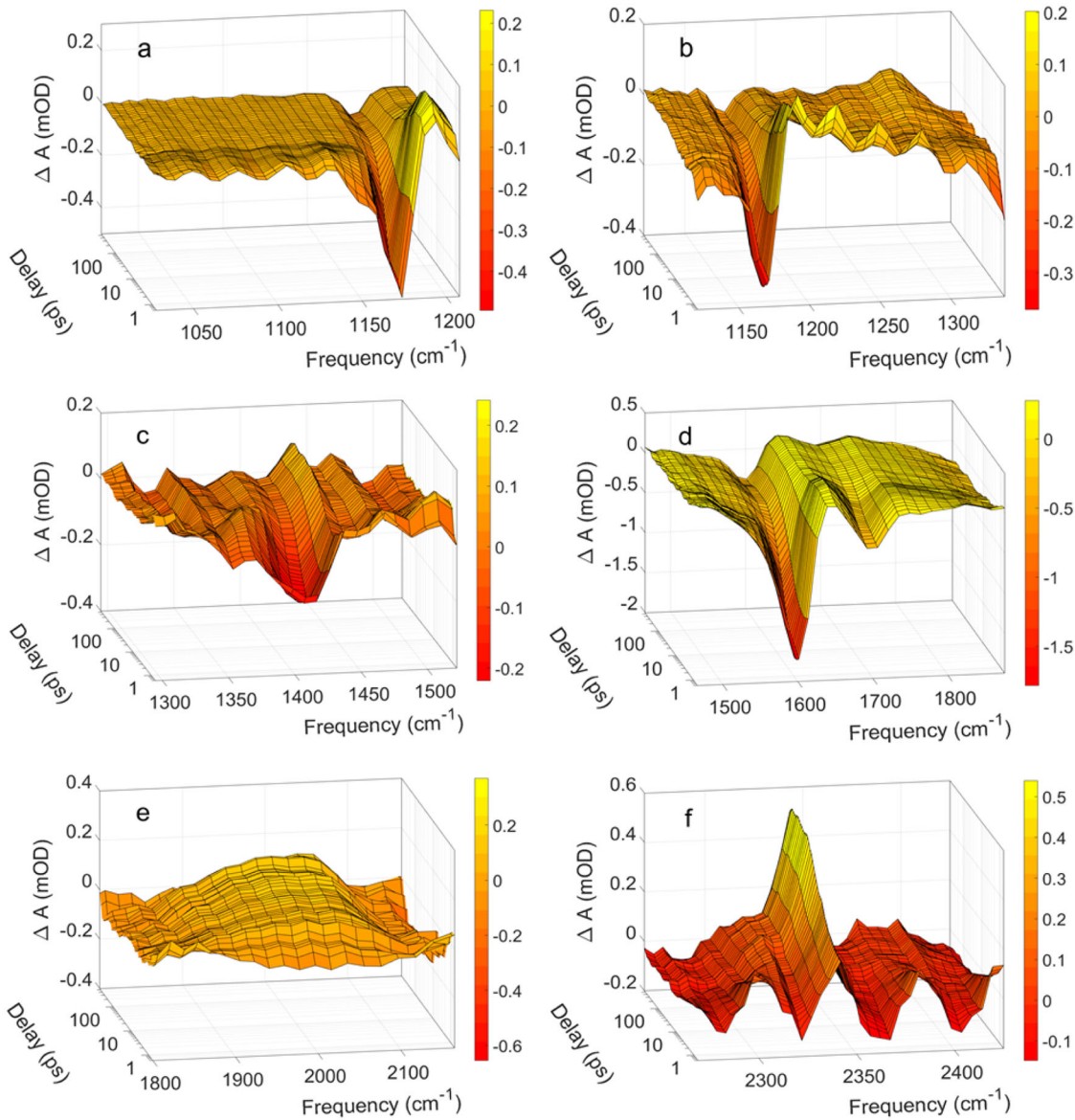

**Fig. 2 | The infrared absorption dynamics of aqueous pyruvate recorded after the 200 nm excitation pulse. a** The negative transient at 1176 cm⁻¹ is associated with the C-CH₃ stretch transition of ground state pyruvate, while the short-lived positive absorption at 1194 cm⁻¹ is assigned to the C-CH₃ stretch transition of excited state pyruvate. **b** The positive absorption appearing at 1295 cm⁻¹ at long delays is assigned to acetic acid. **c** The negative absorption at 1356 cm⁻¹ is assigned to the of the CH₃ umbrella transition of ground state pyruvate. The excitation of the symmetric COO⁻ stretch and symmetric CH₂ bend vibrations of ground state pyruvate merge with the C−COO⁻ stretch transition of ground state pyruvate to give the wide negative absorption with minimum at 1418 cm⁻¹. The positive absorption appearing at 1438 cm⁻¹ at long delays is assigned to acetic acid. **d** The negative absorption associated with the excitation of the asymmetric COO⁻ stretch and the

$C = O$ stretch vibrations of ground state pyruvate are observed at 1603 cm⁻¹ at 1707 cm⁻¹, respectively. The short-lived positive absorption observed around 1666 cm⁻¹ is assigned to excitation of the COO⁻ stretch transition of excited state pyruvate. The positive absorption appearing at 1735 cm⁻¹ at long delays is assigned to acetic acid. **e** The wide positive absorption with maximum around 2000 cm⁻¹ reflects the spectral shift of the H₂O combination band induced by the excitation pulse. **f** The absorption at 2341 cm⁻¹ is assigned to the formation of aqueous carbon dioxide. The transient spectra in (**a**–**c**, **f**) are recorded in H₂O while the data in (**d**) is recorded in D₂O. The transient spectra cannot be compared on a common scale. 2D contour plots of the data are shown in Supplementary Fig. 1. Source data are provided as a Source Data file.

despite the high excitation energy of 6.2 eV, only one in five excited pyruvate anions dissociates, while the rest return to the ground state. The error on the quantum yield mainly reflects the uncertainty of the solvent background subtraction and an estimate of how much the absorption associated with the transient feature at 1194 cm⁻¹ (later to be assigned to excited-state pyruvate) contributes to the early absorption at 1176 cm⁻¹.

### Excited state pyruvate following excitation at 200 nm
In addition to the transients pertaining to vibrational relaxation of ground state pyruvate, Fig. 2 shows two positive, short-lived transients:

one at 1194 ± 20 cm⁻¹ and one at 1666 ± 30 cm⁻¹. The exact frequencies of these absorption transients are uncertain, since they partially overlap the strong negative absorption associated with ground state pyruvate. Figure 5b shows the transient absorption at 1194 cm⁻¹ as a function of time. The absorption trace is well approximated by a single exponential function with a time constant of 9.2 ± 0.3 ps, but the exact time dependence may be perturbed by the nearby negative transients. The decay time of the 1194 cm⁻¹ transient is comparable to the time constant for the ground state absorption recovery suggesting that the short-lived transient is association with excited state pyruvate. The calculated absorption spectrum of the lowest excited singlet and

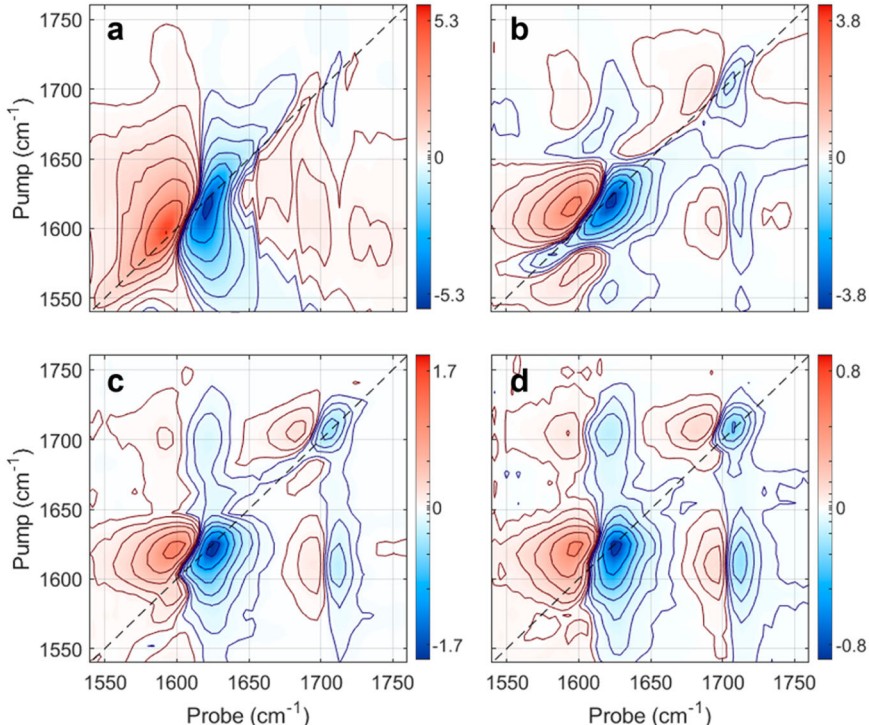

**Fig. 3 | 2D-IR spectra of pyruvate dissolved in D$_2$O. Delays. a** 0 fs, (**b**) 500 fs, (**c**) 1000 fs, and (**d**) 1500 fs. The color bars represent the signal in arbitrary units that allow comparison between the spectra. The color scale is linear but the contours are nonlinearly spaced from 1% to 90% intensity. Source data are provided as a Source Data file.

triplet states of pyruvate coordinated by two H$_2$O molecules is depicted in Fig.6. Replacing the H$_2$O molecules with D$_2$O does not change the absorption spectra significantly. The calculated spectra are obtained using the ωB97X-D density functional method[20] with the aug-pcseg-1 basis set[21], including anharmonic corrections, and the IEFPCM water solvent model, as implemented in the Gaussian16 program package[22]. We have included two explicit water molecules to provide hydrogen bonding to the carbonyl and carboxyl groups. Based on calibration studies, the relative absorption intensities are conservatively estimated to be accurate within a factor of two[23–25]. The calculated spectra of both the excited singlet and the excited triplet state have three strong transitions. Considering the uncertainties of the calculated and measured transition frequencies, the frequencies of the two strongest transitions around 1200 cm$^{-1}$ and 1580 cm$^{-1}$ are in fair agreement with those of the short-lived transients at ~1194 cm$^{-1}$ and ~1666 cm$^{-1}$. The weakest of the three calculated transitions around 1030 cm$^{-1}$ is on the edge of the recorded spectral range and not observed experimentally (Fig. 2a.). We tentatively assign the short-lived transients to excited state pyruvate. Note, however, that the uncertainty of the calculated transition frequencies prevents us from determining if the excited state is of singlet or triplet character. The observed spectrum could possibly arise from a combination of the two states.

### The formation of CO$_2$ following excitation at 200 nm

In addition to the positive absorption transients assigned to excited state pyruvate and vibrationally excited pyruvate in the electronic ground state, the data in Fig. 2 includes a few positive transients persisting for the duration of the measurements. Most notably, the transient absorption data in Fig. 2f shows the asymmetric stretch transition of aqueous CO$_2$ at 2341 cm$^{-1}$ [26–28]. This absorption transient indicates that a fraction of the excited pyruvate anions dissociate by decarboxylation. The early absorption dynamics in Fig. 2f is somewhat hidden by the oscillating solvent signal. Figure 7a shows the same spectral range recorded with five times the pump pulse energy used in

the measurements displayed in Fig. 2. The absorption dynamics of Fig. 7a, f are nearly identical, but the oscillating solvent contribution is absent in Fig. 7a thus giving a better view of the early CO$_2$(aq) formation. The CO$_2$(aq) transient is present already after 1 ps and increases to a constant level after about 50 ps. The absorption trace at 2341 cm$^{-1}$ shown in Fig. 7b is well described by a double exponential function with time constants of $\tau_1 = 1.7 \pm 0.3$ ps and $\tau_2 = 13.9 \pm 0.6$ ps. The fast component, $\tau_1$ is four times faster than the 6.5 ps time constant for the return of pyruvate to its ground state implying that the dissociation of CO$_2$ occurs from an excited state of pyruvate.

Upon dissociation from pyruvate the geometry of the CO$_2$ fragment is transformed from its bent geometry of the carboxylate to the linear CO$_2$(aq) geometry and the concurrent vibrational relaxation therefore likely occurs in the bending mode. We are unable to follow this relaxation as the bending transition frequencies of CO$_2$ are outside the range of our transient absorption spectrometer. However, the transition associated with the $v_3 = 1 \to 2$ hot band of the CO$_2$(aq) asymmetric stretch at 2314 cm$^{-1}$, and the $v_3 = 2 \to 3$ hot band of the CO$_2$(aq) asymmetric stretch at 2304 cm$^{-1}$ [26–28] are only barely visible in Fig. 7a, consistent with the relaxation predominantly taking place in the bending mode.

The vibrational relaxation of aqueous CO$_2$ following excitation of its fundamental ($v_3 = 0 \to 1$) asymmetric stretch transition has been studied across a wide temperature range by Gleim et al.[29]. The studies showed that the relaxation time of the fundamental asymmetric stretch increases with temperature from $\tau = 8.9$ ps at $T = 303$ K to $\tau = 16.3$ ps at 603 K. The relaxation times are consistent with the $\tau_2 = 13.9$ ps time scale for CO$_2$(aq)'s absorption dynamics in Fig. 2f and Fig. 7 and indicate that the late development of CO$_2$(aq)'s absorption after decarboxylation of pyruvate is at least in part due to vibrational relaxation. Accordingly, CO$_2$ is likely formed by dissociation from an electronic excited state of pyruvate in 1.7 ps and the increasing absorption at 2341 cm$^{-1}$ results from vibrational relaxation of CO$_2$(aq). We have previously observed the same absorption dynamics of CO$_2$(aq)'s asymmetric stretch transition following the photo-

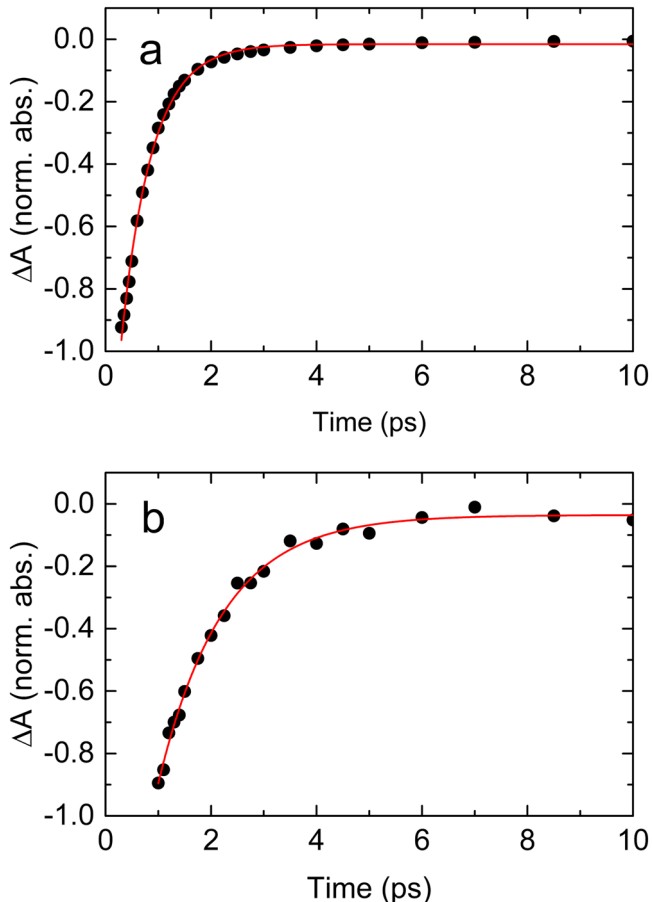

**Fig. 4 | Vibrational relaxation dynamics of fundamental ground state transitions of pyruvate. a** Relaxation dynamics of the fundamental asymmetric COO⁻ and (**b**) the CO Stretch represented by the negative components of their diagonal signals (dots) from Fig.3. The absorption dynamics of the asymmetric COO⁻ stretch is well approximated by the single exponential function: $\Delta A = -1.950 \times \exp(-t/0.6\,ps) -0.016$ (red line). The absorption dynamics of the CO stretch is well approximated by the single exponential function: $\Delta A = -1.60 \times \exp(-t/1.2\,ps) -0.036$ (red line). Source data are provided as a Source Data file.

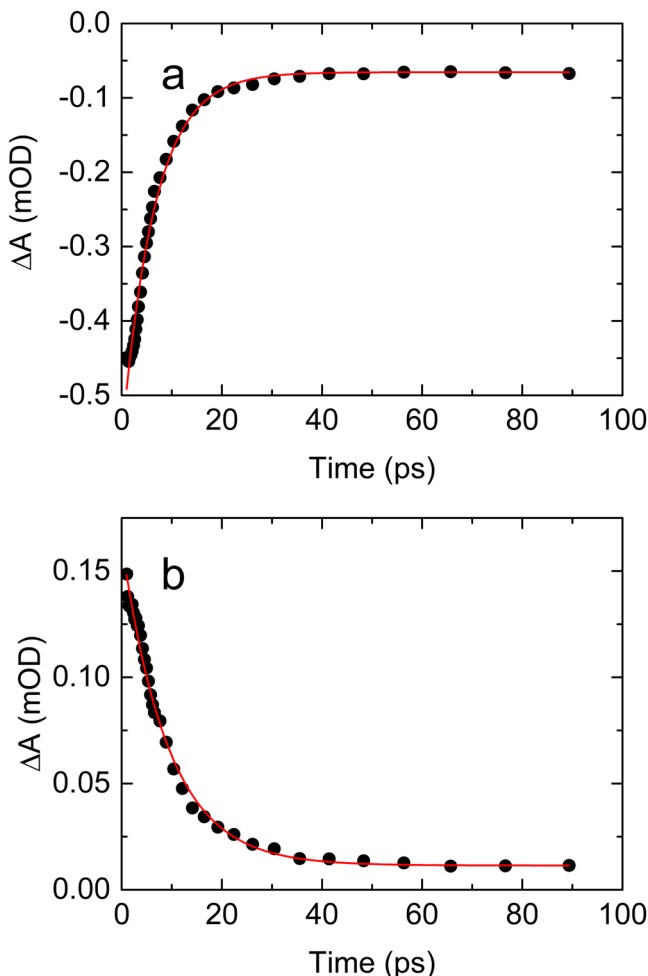

**Fig. 5 | Transient absorption recovery of ground state pyruvate after excitation at 200 nm. a** The transient absorption recovery of the C-CH3 stretch at $1176\,cm^{-1}$ (dots). The absorption dynamics is well approximated by the single exponential function: $\Delta A = -0.50\,mOD \times \exp(-t/6.5\,ps) -0.065\,mOD$ (red line). **b** The transient absorption trace at $1194\,cm^{-1}$ (dots). The absorption dynamics is well approximated by the single exponential function: $\Delta A = 0.15\,mOD \times \exp(-t/9.2\,ps) + 0.011\,mOD$ (red line). Source data are provided as a Source Data file.

decarboxylation of a number of carboxylates, amino acid zwitterions and di-peptides suggesting a common $CO_2$ vibrational relaxation path for these molecules[30–34] .The relatively slow relaxation of $CO_2$ likely indicates a weak interaction between $CO_2$ and its hydration shell[35]. As our spectrometer is unable to cover the positive $CO_2$(aq) absorption transient at $2341\,cm^{-1}$ simultaneously with any of the negative absorption transients associated with the excitation of pyruvate in one measurement, we are unable to determine the quantum yield of $CO_2$(aq).

In addition to the absorption pertaining to $CO_2$(aq), Fig. 2 shows three positive absorption peaks at $1295\,cm^{-1}$, $1438\,cm^{-1}$ and $1735\,cm^{-1}$ persisting for the duration of the measurements. The positive peaks imply the formation of new species. The formation of $CO_2$ implies the production of the acetyl anion, $CH_3CO^-$, as the residual species. The calculated spectrum of $CH_3CO^-$, however, is incompatible with the observed spectrum, and the calculated spectra of acetyl radical, $CH_3CO^{\cdot}$, acetaldehyde, $CH_3CHO$, or acetaldehyde diol $CH_3C(OH)_2$ do not match the observed transitions neither. However, our calculations indicate that the acetyl anion readily abstracts a hydroxyl radical from water while releasing an electron, thereby forming acetic acid. The infrared spectrum of acetic acid in $H_2O$, shown in Fig. 7c, has three dominating peaks at $1278\,cm^{-1}$, $1390\,cm^{-1}$ and $1713\,cm^{-1}$. Considering that the transient peaks observed at $1438\,cm^{-1}$, $1735\,cm^{-1}$ are partially hidden by negative absorption transients pertaining to ground state

pyruvate, acetic acid seems a plausible candidate for the persistent absorption transients in Fig. 2. The combined experimental and theoretical results following 200 nm excitation thus suggest the reaction:

$$h\nu_{200nm} + CH_3COCOO^- + OH \rightarrow CH_3COOH + CO_2 + e_{aq}^- \quad (4)$$

Assuming this assignment is correct, we can use the ground state pyruvate transient absorption trace at $1176\,cm^{-1}$ and the acetic acid transient absorption trace at $1278\,cm^{-1}$ together with their corresponding steady state infrared absorption spectra to assess the quantum yield of acetic acid. We estimate that $Y(t=100\,ps) = 50\% \pm 20\%$ of the pyruvate anions that remain excited after 100 ps have formed acetic acid after 100 ps. After 560 ps the yield is $Y(t=560\,ps) = 84\% \pm 30\%$. Consequently, following the excitation of aqueous pyruvate at 200 nm, the dominating fraction, if not all, of the dissociating molecules dissociate according to reaction (4).

## Transient infrared absorption dynamics of pyruvate after excitation at 340 nm
We have measured the transient infrared absorption spectra $\Delta A(\nu,t)$ between $1030\,cm^{-1}$ and $2420\,cm^{-1}$ following the photoexcitation of aqueous pyruvate at 340 nm. Due to the small extinction coefficient of

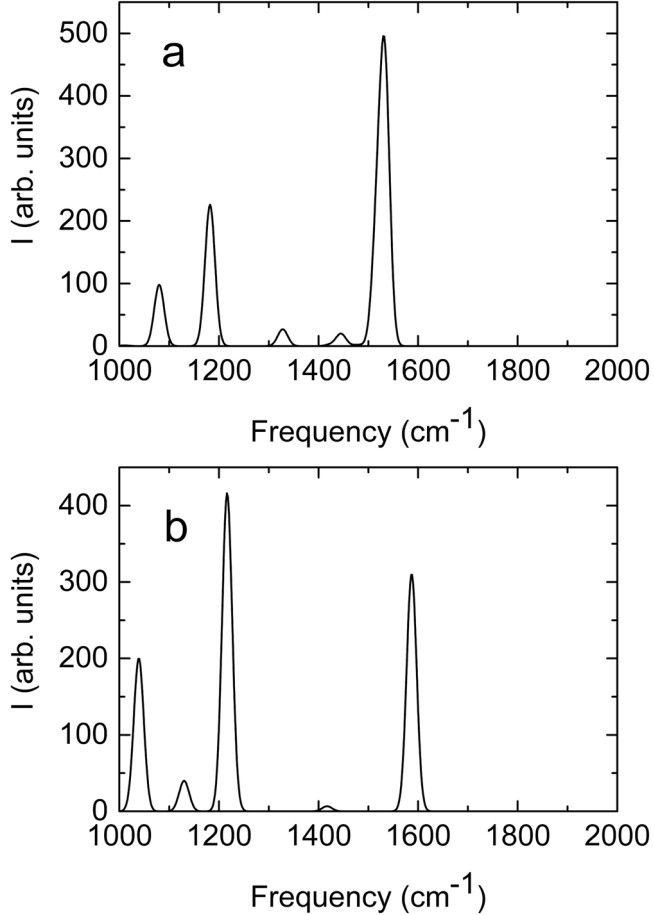

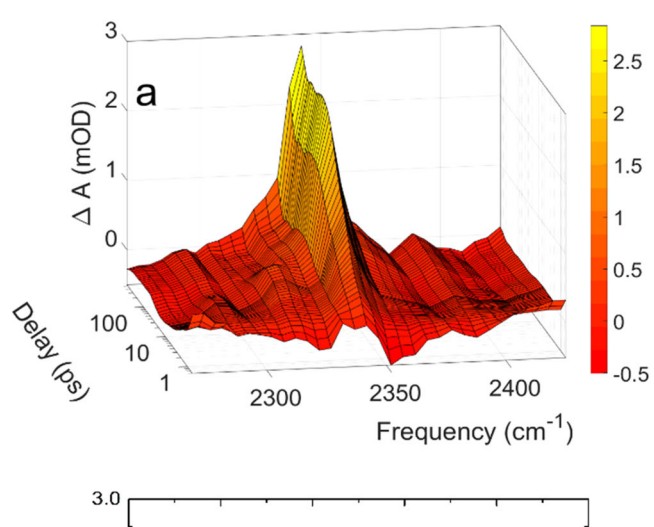

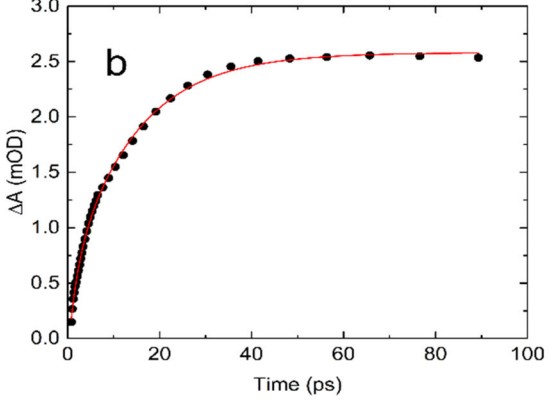

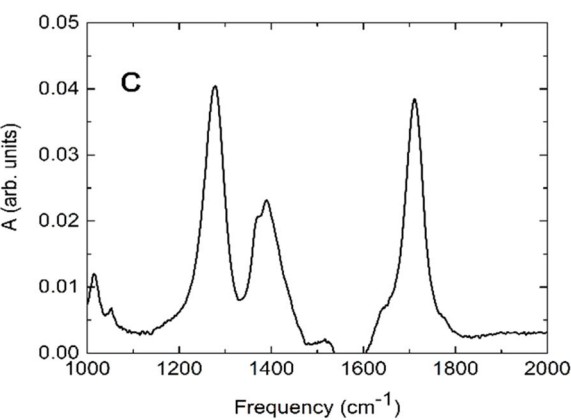

**Fig. 6 | Calculated absorption spectra of excited state pyruvate. a** Lowest singlet excited state and (**b**) the lowest triplet excited state of pyruvate. The spectral width is arbitrarily set to 25 cm⁻¹ (FWHM) for clarity. Source data are provided as a Source Data file.

**Fig. 7 | Photoproducts from aqueous pyruvate following excitation at 200 nm. a** Transient absorption dynamics showing the formation of $CO_2(aq)$. 2D contour plot of Fig. 7a is shown in Supplementary Fig. 2. **b** Transient absorption trace recorded at 2341 cm⁻¹ (dots). The transient 2341 cm⁻¹ is well approximated by the function: $\Delta A = -0.67$ mOD$\times \exp(-t/1.7$ ps$) -2.0 \times \exp(-t/13.9$ ps$) + 2.58$ mOD (red line). **c** Steady state infrared absorption spectrum of acetic acid recorded in $H_2O$ against a $H_2O$ reference. Source data are provided as a Source Data file.

aqueous pyruvate at 340 nm, only data in the ranges 1090 –1290 cm⁻¹ and 1490 –1880 cm⁻¹ show transient absorption signals above the noise floor. Specifically, in spite of repeated measurements with excellent statistics in the spectral region of the asymmetric stretch of $CO_2(aq)$, we did not identify this species. Figure 8a and Fig. 8b show the strongest transitions of ground state pyruvate as negative induced absorption peaks: The C-$CH_3$ stretch transition at 1176 cm⁻¹, the asymmetric $COO^-$ stretch transition at 1603 cm⁻¹ and the C = O stretch at 1707 cm⁻¹. The negative absorption transients drop to a minimum immediately after the excitation pulse and then fully recover to their initial value after 200 ps. As argued above, the transient absorption spectra reflect the ground state pyruvate concentration dynamics. The quantum yield for photodissociation of pyruvate at 340 nm can therefore be calculated from the transient absorption data. Figure 8c shows the ground state recovery represented by the absorption trace at 1176 cm⁻¹. The measured data are well approximated by the exponential function $\Delta A(t) = -0.131$ mOD $\times \exp(-t/13.1$ ps$) - 0.003$ mOD. The ground state concentration thus recovers on a $13.1 \pm 0.8$ ps time scale, which is twice as long as the ground state recovery time after excitation at 200 nm. Using the average transient absorption after 200 ps, we find a primary photo-dissociation quantum yield for the 340 nm excitation of $\Phi(t \geq 200$ ps$) = \Delta A_{av}(t > 200$ ps$)/\Delta A_{av}(t = 1$ ps$) = 0.5\% \pm 3\%$. As for the yield related to the 200 nm excitation, the error on the quantum yield following excitation at 340 nm reflects the uncertainty of the solvent background subtraction and an estimate of how much the absorption associated with the transient feature with

maximum at 1204 cm⁻¹ contributes to the early absorption at 1176 cm⁻¹.

In addition to the negative ground state absorption, Fig. 8a shows a positive induced absorption with maximum at 1204 cm⁻¹. The positive absorption is represented by the absorption trace at 1204 cm⁻¹ in Fig. 8d and is well approximated by a single exponential function $\Delta A(t) = 0.065$ mOD$\times \exp(-t/12.4$ ps$) + 0.004$ mOD. The positive absorption thus decays on a $12.4 \pm 0.8$ ps timescale. Within the experimental uncertainty, this time scale is identical to that of the

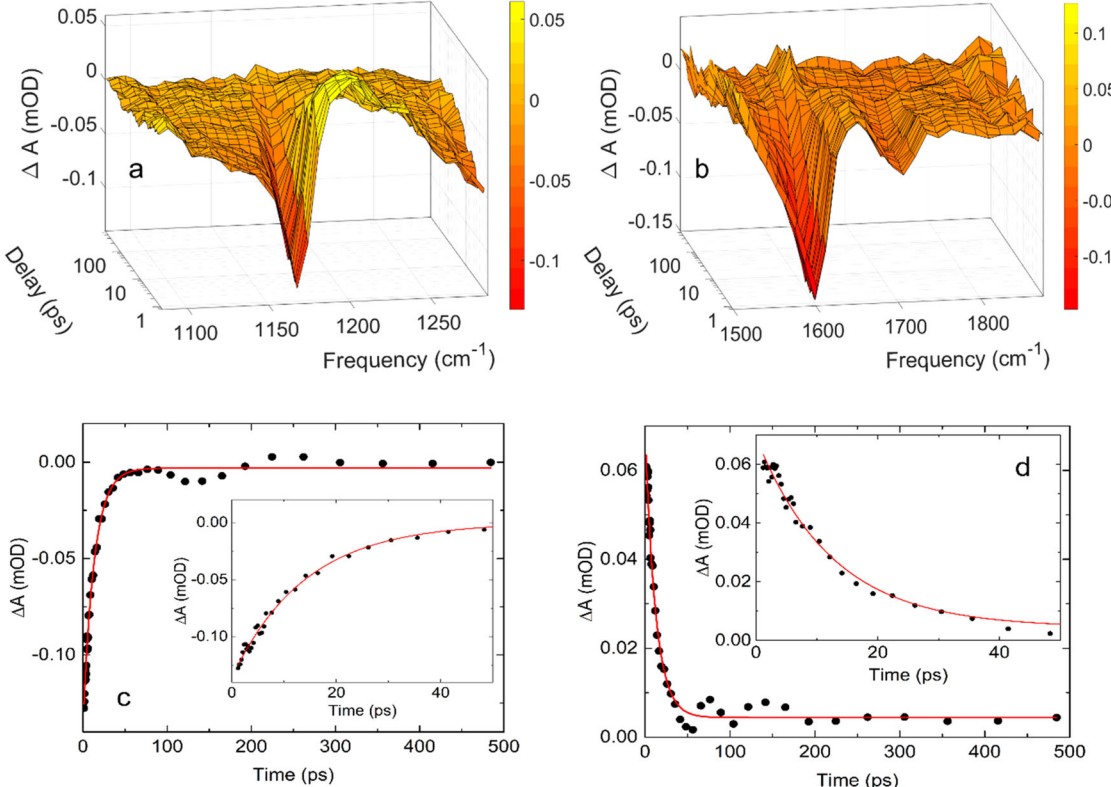

**Fig. 8 | The infrared absorption dynamics following photoexcitation of aqueous pyruvate at 340 nm. a** 0.3 M pyruvate in $H_2O$. **b** 0.1 M pyruvate in $D_2O$. 2D contour plots of Fig. 8a and Fig. 8b are shown in Supplementary Fig. 3. **c** Transient absorption trace at 1176 cm$^{-1}$, and (**d**) transient absorption trace at 1204 cm$^{-1}$. The measured data in **c** is approximated with the function $\Delta A = -0.135$ mOD exp($-t/$ 13.1 ps) $- 0.003$ mOD (red curve), while the measurements in (**d**) are approximated with the function $\Delta A = 0.065$ mOD exp($-t/12.4$ ps) $+ 0.005$ mOD (red curve). The insets show the first 50 ps after the 340 nm excitation pulse. Source data are provided as a Source Data file.

ground state absorption recovery, indicating that the species responsible for the absorption at 1204 cm$^{-1}$ is returning to the pyruvate ground state. Though significantly slower, the spectral dynamics of the 1204 cm$^{-1}$ transient induced by 340 nm excitation is like the short-lived absorption observed in the same spectral region after excitation at 200 nm. This absorption was assigned to excited state pyruvate above and the measurements shown in Fig. 8a, d confirm this assignment. Population of the second excited state by 200 nm radiation is expected to decay to the lowest excited state within 1 ps[36], and pyruvate anions excited at 200 nm and 340 nm therefore likely populate the same lowest excited state before returning to the ground state. The higher excess energy following excitation at 200 nm allows the excited pyruvate anions to access a larger part of the excited state potential surface thereby increasing the chance of entering a faster decay path to the ground state than pyruvate anions excited at 340 nm. This raises the important question of why excitation of aqueous pyruvate at 200 nm leads to decarboxylation, whereas excitation at 340 nm does not. The answer is found by energy considerations: Fig. 9 shows calculated (ωB97X-D/aug-pcseg-2 single points at ωB97X-D/aug-pcseg-1 geometries) energy profiles as a function of the C(=O)-CO$_2$ bond length for pyruvate solvated by two water molecules, with the black curve representing the relaxed ground state potential, the red and blue dashed curves representing the singlet and triplet vertical excited states, and the red and blue solid curves representing the lowest geometry relaxed singlet and triplet states, respectively. The geometry corresponding to the relaxed excited singlet state at a distance of 2.4 Å has an excitation energy of only 1.8 eV. At longer distances the ground and excited states become so close in energy that attempts of geometry optimization fail, as predicted geometry steps lead to state switching. The employed TDDFT method cannot describe regions with

near-degeneracy between the ground and first excited states, and the observed behavior can thus only be taken as indirect evidence of either a conical intersection or an accidental near-degeneracy of the ground and excited states. Excitation at 340 nm (3.6 eV) has just enough energy to reach the lowest singlet excited state at the ground state equilibrium structure, but this leaves insufficient energy to also overcome the ~1 eV barrier on the relaxed singlet excited surface. In contrast, excitation at 200 nm (6.2 eV) to the second singlet excited state is expected to rapidly decay to the lowest excited state by internal conversion[36], and this leaves ~2.5 eV as internal kinetic energy, which is more than sufficient to lead to dissociation of CO$_2$. Note that there is virtually no barrier on the relaxed triplet excited surface. Since the photo-dissociation quantum yield of pyruvate upon excitation at 340 nm is insignificant, this suggests that intersystem crossing from the singlet to the triplet excited state never takes place, likely because much faster processes such as internal conversion to the ground state outcompetes intersystem crossings.

The present study finds that photoexcitation of pyruvate with deep-UV irradiation at 200 nm leads to decarboxylation yielding carbon dioxide and acetic acid. About one out of five excitations result in decarboxylation, while the remaining excited pyruvate anions return to their native ground state. The observation of decarboxylation at 200 nm excitation is somewhat anticipated, given the known behavior of carboxylates when subjected to deep-UV irradiation. Nevertheless, this study not only reaffirms this decarboxylation mechanism and measures the yield, but also provides a robust foundation upon which we can predict the subsequent secondary reactions.

Near-UV excitation of aqueous pyruvate at 340 nm, on the other hand, is insufficient to induce photochemical reactions, and all excited molecules return to the native ground state after having spent about

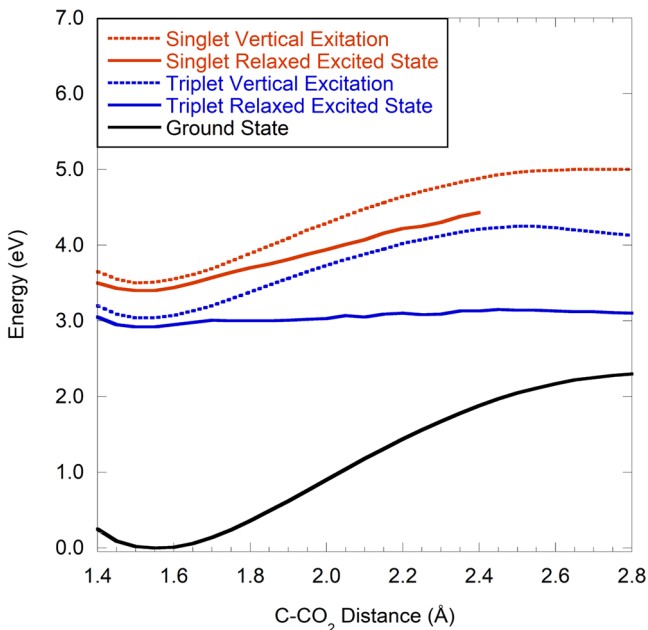

**Fig. 9 | Calculated potential curves for ground state and first excited states of pyruvate with two water molecules.** Source data are provided as a Source Data file.

13 ps in the singlet excited state. Accordingly, solar photoexcitation of pyruvate in aerosols produces little if any photoproducts. These findings are in line with the closing of the photodissociation channels of aqueous pyruvate clusters observed when two or more water molecules are added to the pyruvate anion[13]. Furthermore, the present study of aqueous pyruvate also explains the observed drop in photodissociation yield of pyruvic acid with increasing pH.

These mechanistic insights may help understand the role of pyruvate photochemistry in atmospheric processes with implications for both theoretical modeling and empirical observation in atmospheric science.

## Methods
### Experimental details
**Femtosecond transient absorption spectroscopy.** We use femtosecond UV pump-IR probe transient absorption spectroscopy to study the primary photolysis of the aqueous pyruvate following excitation at $\lambda = 200$ nm and $\lambda = 340$ nm. The beam of 200 nm pump pulses used for exciting the sample is generated by frequency quadrupling a beam of 800 nm femtosecond laser pulses from an amplified Titanium:Sapphire laser in three consecutive β-barium borate crystals. The beam of 340 nm pump pulses is generated by frequency quadrupling the output from an optical parametric amplifier pumped by a beam of 800 nm femtosecond laser pulses from an amplified Titanium:Sapphire laser in two consecutive β-barium borate crystals. The pump pulses are sent via a scanning delay line and through a half-wave plate before they are focused through the sample by a concave mirror. A mechanical chopper modulates the beam of pump pulses such that every second pump pulse excites the sample, while the rest are blocked. The beam of infrared probe pulses is generated by difference frequency mixing the signal and idler pulses from an optical parametric amplifier pumped by the amplified Titanium:Sapphire laser. The beam of probe pulses is divided into a signal and reference beam by a beam splitter. The pulses in the signal beam probes the sample inside the volume excited by the pump pulses, while the reference beam passes through the sample outside the volume excited by the pump beam. The probe pulses are subsequently analyzed and detected by a spectrometer equipped with a dual array HgCdTe detector. The spectrum of the signal pulses is normalized to that of the reference pulses and the transient induced absorption spectrum is obtained by subtracting the normalized probe spectrum recorded with pump pulse excitation from the normalized probe spectrum recorded without pump pulse excitation.

The samples consist of a constantly flowing wire guided film of aqueous sodium pyruvate. The flow ensures a fresh sample for every probe pulse and frequent replacement of the sample minimizes the buildup of permanent photoproducts. The transient absorption measurements use $H_2O$ as solvent except for measurements conducted in the spectral range 1550–1750 cm$^{-1}$, where the absorption associated with the $H_2O$ bending mode renders the sample opaque. Instead $D_2O$ is used as solvent in this range. All measurements are recorded with the pump and probe polarizations at the magic angle of 54.7°. The pyruvate samples have a concentration of 0.30 M. However, due to the strong infrared absorption of the asymmetric carboxylate stretch the data is recorded with a pyruvate concentration of 0.10 M. The acid constant of pyruvate is $pK_a = 2.5$ and with a measured solution pH = 7.3, the pyruvate is on the anionic form shown in Fig. 1. The data measured for $-0.5 < t < 0.5$ ps are obscured by the $t = 0$ ps coherence signal, and the transient absorption spectra therefore only shows data after $t > 0.5$ ps. The oscillating transient spectrum observed during the first picosecond in Fig. 2a, b, f are remnants of the coherent solvent signal. For excitation at 340 nm, the data measured for $-0.1 < t < 0.1$ ps are obscured by the $t = 0$ ps coherence signal, and Fig. 9 therefore only shows data after $t > 0.2$ ps. The spectral resolution of the transient absorption spectrometer is 9 cm$^{-1}$ in Fig. 2a–e and Fig. 9 and 2 cm$^{-1}$ in Figs. 2f, 7a. To suppress the effect of multi-photon ionization of the water molecules to below the detectable level, the excitation energy is kept at 1.0 µJ, for the 200 nm excitation and 3.4 µJ when exciting at 340 nm. The linear pump-induced water solvent background signal has been subtracted from all data.

**Femtosecond 2D-IR spectroscopy.** We furthermore investigate the intra-molecular coupling of the vibrational modes of pyruvate using a commercial 2D-IR spectrometer with the pump-probe geometry from Phasetech. All measurements are performed with the pump and probe beam polarizations at magic angle, $\theta = 54.7°$, to exclude contributions to the absorption dynamics from rotational dipole effects. The Mid-infrared pulses for the 2D-IR spectrometer are generated by difference frequency mixing the infrared pulses from an optical parametric amplifier pumped by a femtosecond Yb:KGW laser, all from Light Conversion. The samples used for these measurements consist of 0.1 M of pyruvate dissolved in $D_2O$. The samples are placed between two CaF$_2$ windows with a spacing of 100 µm.

**Steady-state spectroscopy**
The UV–Vis spectra of aqueous pyruvate are recorded by a Shimadzu UV-3600 spectrometer. The absorbance and wavelength of the Shimadzu spectrometer are calibrated against NaCl(aq) and KCl(aq) certified calibration references (RM-KCSCLCKISI) from Starna. The steady-state IR spectra are recorded by ATR FTIR with the resolution set to $\Delta\nu = 4$ cm$^{-1}$ on a Nicolet 380 spectrometer from Thermo Fisher.

**Computational details**
All calculations have been done using the ωB97X-D functional[20] and the aug-pcseg-1 or aug-pcseg-2 basis sets[21] and the IEFPCM implicit solvent model[22]. The ωB97X-D includes a range-separated X-functional and empirical dispersion, with the former important to accurately describe possible charge-transfer character of the excited states and the latter important for describing the interaction with explicit water molecules. Excitation energies have been calculated within the TDDFT framework, where only the optical part of the dielectric constant in the IEFPCM model is included. The two explicit water molecules are located such that they provide hydrogen bonding to the ketone and carboxyl oxygen atoms. All calculations have been performed using the Gaussian-16 software package[22].

## Data availability

The primary data consists of spectral scans as a function of time delay and are provided in the Source Data file. Source data are provided with this paper.

## Code availability

The data analysis leading to the figures in the paper and supplementary information has been done using home-made code based on Labview 7.1. The code can be obtained upon request.

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

## Acknowledgements

T.W. and J.T. acknowledge the Novo Nordisk Foundation Facility Grant NanoScat No. NNF18OC0032628 for funding, and the Danish National Research Foundation for financial support through the Center of Excellence for Chemistry of Clouds (Grant Agreement No. DNRF172). F.M. acknowledges funding from the European Union's Horizon 2020 Research and Innovation Program under the Marie Skłodowska–Curie Grant No. 101024120.

## Author contributions

J.T. and T.W. designed and directed the project. J.T. recorded and analyzed the stationary and transient absorption data. F.M. recorded and analyzed the 2D-IR data. F.J. provided the theoretical calculations. J.T.

and F.J. interpreted the data and wrote the manuscript with help from F.M. and T.W. F.J. is the corresponding author frj@chem.au.dk.

## Competing interests

The authors declare no competing interests.
