## [Peer Review File · Nature Communications]

Aqueous pyruvate partly dissociates under deep ultraviolet irradiation but is resilient to near ultraviolet excitationReviewer #1 (Remarks to the Author):

The present manuscript reports on spectroscopic experiments (time-resolved IR, 2D IR aiming to unravel the photochemistry of aqueous pyruvate under deep- and near-UV light. This work shows that actinic photoexcitation does not provide enough energy for a photodissociation of aqueous pyruvate, while deep-UV excitation (200nm) leads to decarboxylation but not for the entire pyruvate population. These results contrast with the gas-phase photochemistry of pyruvate, where a complete dissociation of the molecule was observed, and complement another recent work where microsolvation was shown to limit the dissociation processes triggered by light. This work offers an exciting perspective on the aqueous photochemistry of pyruvate and will be of great interest to the atmospheric and spectroscopic community. I would certainly recommend the publication of a revised version of this work in Nature Communications. Below, The authors will find a series of comments that they should address.

My main comments are mostly related to the computational rationale provided in this work.

- 1) I could not find any computational details related to the calculations performed (apologies if I missed them). Which software was used? While no information could be found in the text, I suppose that TDDFT was utilized for the excited states. Were the calculations using the Tamm-Dancoff approximation? What is the rationale for using the wB97XD functional for this system? Did the authors use an IEFPCM for the TDDFT energy profiles, and if yes, did the authors employ an equilibrium or a non-equilibrium formalism? How were the water molecules placed around the pyruvate? (I suspect a few low-energy configurations are possible for pyruvate + 2 water molecules.) What is the ionization energy of pyruvate? I would guess it should not be far from the electronic states considered. If so, how does that affect the TDDFT calculations? Were the calculations free from any artificial transitions brought about by the (potentially) low ionization potential?
- 2) The authors rationalize their experimental findings using TDDFT energy profiles (p.14). How confident are the authors that TDDFT can adequately describe the CO₂ release (given the limitations of this method for bond dissociations)? Did they benchmark this technique? The authors mention an avoided crossing for the excited singlet state (p.14): what is the character of the electronic state interacting with this singlet state of interest?
- 3) Page 14: what is the indication that the molecule (initially in S₂) will go to S₁ and then follow from there the S₁ energy profile presented? The molecule in S₂ may have its own decay toward another part of the S₁ (or S₁/S₀) energy surface (which perhaps is connected to avoided crossing with S₁ mentioned by the authors). What is the character of S₂?
- 4) On page 9, the authors estimate that the relative absorption intensities are accurate within a factor two. Can the authors explain their rationale for that statement?
- 5) The 3D plots (plots in Fig. 2, Fig. 7a, plots in Fig. 9) would be way more readable and simpler to analyze by the reader if they were presented as 2D colormaps using a uniform color scheme like viridis (see 10.1038/s41467-020-19160-7).

Reviewer #2 (Remarks to the Author):

This is a fine piece of work that studies aqueous pyruvate photochemistry dynamics at deep- as well as in near-UV. The authors made a convincing argument to justify the importance of this work under the context of the Earth's atmosphere and the origin of life. This work also completes our understanding of the photochemistry of pyruvate in gas-, cluster- and aqueous phase in the near-UV region, and extends to the deep-UV. The different photochemistry behaviors were reported at two excitations and quantum yields for certain products (e.g. CO₂ (aq)) were estimated. Both experiments and theoretical modellings were competently and carefully conducted. I only have a few minor quires for the authors: 1) on page 13, the quantum yield of photo-dissociation is given 0.5 +/-3%. It could be misunderstood as 50%. It would be better to express (0.5 +/-3)%; 2) page 15, the reference is 13 not 12 when referring to two water molecules blocking all dissociation channels.

Manuscript ID: NCOMMS-23-54100

Title: "Aqueous pyruvate partly dissociates when exposed to deep ultraviolet irradiation, but is resilient to near ultraviolet excitation"

Author(s): Jan Thøgersen, Fani Madzharova, Tobias Weidner and Frank Jensen

I have uploaded a revised version of the above manuscript. The revision has been done according to the referee comments as indicated below.

I hope the revised manuscript can be considered for publication in Nature Comm.

Sincerely,
Frank Jensen

REVIEWER COMMENTS

Reviewer #1 (Remarks to the Author):

The present manuscript reports on spectroscopic experiments (time-resolved IR, 2D IR aiming to unravel the photochemistry of aqueous pyruvate under deep- and near-UV light. This work shows that actinic photoexcitation does not provide enough energy for a photodissociation of aqueous pyruvate, while deep-UV excitation (200nm) leads to decarboxylation but not for the entire pyruvate population. These results contrast with the gas-phase photochemistry of pyruvate, where a complete dissociation of the molecule was observed, and complement another recent work where microsolvation was shown to limit the dissociation processes triggered by light.

This work offers an exciting perspective on the aqueous photochemistry of pyruvate and will be of great interest to the atmospheric and spectroscopic community. I would certainly recommend the publication of a revised version of this work in Nature Communications. Below, The authors will find a series of comments that they should address.

My main comments are mostly related to the computational rationale provided in this work.

1) I could not find any computational details related to the calculations performed (apologies if I missed them). Which software was used? While no information could be found in the text, I suppose that TDDFT was utilized for the excited states. Were the calculations using the Tamm-Dancoff approximation? What is the rationale for using the wB97XD functional for this system? Did the authors use an IEFPCM for the TDDFT energy profiles, and if yes, did the authors employ an equilibrium or a non-equilibrium formalism? How were the water molecules placed around the pyruvate? (I suspect a few low-energy configurations are possible for pyruvate + 2 water molecules.) What is the ionization energy of pyruvate? I would guess it should not be far from the electronic states considered. If so, how does that affect the TDDFT calculations? Were the calculations free from any artificial transitions brought about by the (potentially) low ionization potential?

All calculations used the Gaussian-16 software package. The TDDFT did not use the TD approximation, as this leads to marginal savings, and the TDDFT label is always taken to be the full response calculation, unless explicitly stated otherwise. The wB97X-D was selected as a functional including both range-separation and dispersion, the former important for describing possible charge-transfer character in the excited states, the latter important for accurately describing the interaction with explicit water molecules. All calculations used the IEFPCM implicit solvent model, with the full dielectric constant for

geometry optimizations on the ground state, but only the optical component for the excitation energies. The two explicit water molecules are located such that one provides hydrogen bonding to the ketone oxygen and one of the carboxyl oxygen while the other provides hydrogen bonding to both the carboxyl oxygens. The exact structure was determined by geometry optimization. The calculated IP of pyruvate with the continuum solvent model is 6.3 eV, the corresponding value with two explicit water molecules is 6.7 eV. The calculated gas-phase value is 3.8 eV, which can be compared with the experimental values of 4.04 ± 0.18 eV. The calculated TDDFT excitation energies 4.2 and 5.8 eV can be compared with the corresponding EOMCCSD values of 4.3 and 5.9 eV. The TDDFT excitation energies with two explicit water molecules are 3.9 and 5.2 eV. These results suggest that the IP is ~1.5 eV higher than the S2 state, and no interference with the excited states is thus expected or observed. We have added the following in the Methods section, with appropriate references. “All calculations have been done using the ωB97X-D functional and the aug-pcseg-1 basis set and the IEFPCM implicit solvent model. The ωB97X-D includes a range-separated X-functional and empirical dispersion, with the former important to accurately describe possible charge-transfer character of the excited states and the latter important for describing the interaction with explicit water molecules. Excitation energies have been calculated within the TDDFT framework, where only the optical part of the dielectric constant is included. The two explicit water molecules are located such that they provide hydrogen bonding to the ketone and carboxyl oxygen atoms. All calculations have been performed using the Gaussian-16 software package.”

2) The authors rationalize their experimental findings using TDDFT energy profiles (p.14). How confident are the authors that TDDFT can adequately describe the CO2 release (given the limitations of this method for bond dissociations)? Did they benchmark this technique? The authors mention an avoided crossing for the excited singlet state (p.14): what is the character of the electronic state interacting with this singlet state of interest?

The dissociation of CO2 from pyruvate leads to two closed shell molecules, and a singlet determinant wave function should thus be valid for the ground state surface. The TDDFT produces the excited state surface by adding the vertical excitation energy, and to the extent that the TDDFT excitation energy is accurate, the excited state surface should thus be accurate to the same degree. The calibration of the excitation energies against EOMCCSD, and the calibration of the IP against experiments, suggests that the employed methodology is valid. We have not performed calibration studies, as this would require CAS-MR2 type calculations with quite large CAS spaces, and this would be a computationally significant effort.

Avoided crossings cannot be described by TDDFT and actually locating such a point is thus not possible within the employed method. The indirect signature is that the lowest calculated excitation energy becomes very close to zero, and the avoided crossing is thus with the ground state. In the context of a constrained optimization, as in figure 11, the geometry step predicted by the gradient for the lowest excited state leads to a geometry where the ground state becomes the excited state, and the gradient at the new geometry thus belongs to a different state. This will predict a new geometry that may have yet another switch in energetic order. A sequence of such geometry steps and associated gradients on alternating energy surfaces leads to non-convergence, as optimization algorithms assume a continuous energy surface. Note that figure 11 does not reflect the avoided crossing with the ground state, as this is a one-dimensional cut of the energy surfaces with all other degrees of freedom relaxed for the individual states. Thus, the avoided crossing of the lowest excited singlet state near 2.4 Å with the ground state is for the excited state relaxed geometry, while the black curve is the energy for the ground state relaxed geometry. We have made it clear in the revised manuscript that the avoided crossing is with the ground state by the following sentence “The relaxed excited singlet state has an avoided crossing with the

ground state at a distance of 2.4 Å, inferred by a near-zero excitation energy, preventing geometry optimizations for longer distances.”

3) Page 14: what is the indication that the molecule (initially in S2) will go to S1 and then follow from there the S1 energy profile presented? The molecule in S2 may have its own decay toward another part of the S1 (or S1/S0) energy surface (which perhaps is connected to avoided crossing with S1 mentioned by the authors). What is the character of S2?

Kasha's rule states that internal conversion from S2 to S1 in essentially all systems is very fast, as only the S1 displays fluorescence. The two main exceptions are azulenes and thioketones, where the S2 state has a lifetime approaching 1 ns. For standard organic molecules, however, the S2 lifetime is substantially less than 1 ps (Russ. Chem. Rev. 70, 471). As stated in the original manuscript, the S1 state can in the perpendicular ground state geometry be considered as a localized n-pi* on the ketone, while the S2 can be considered as a localized n-pi* on the carboxyl group. In a planar geometry, these two states mix substantially, and since the rotational energy barrier is low (few kJ/mol), the experimental results are a Boltzmann average of such mixed states. We believe that this facile mixing of the two states will lead to a rapid S2 to S1 conversion, likely within 1 ps, and thus all important dynamics occur on the S1 surface. We have modified the sentence in the original manuscript: “Thus, the pyruvate anions excited at 200 nm and 340 nm likely populate the same lowest excited state before returning to the ground state.” To “Population of the second excited state by 200 nm radiation is expected to decay to the lowest excited state within 1 picosec, and pyruvate anions excited at 200 nm and 340 nm therefore likely populate the same lowest excited state before returning to the ground state.” to reflect this and included the *Russ. Chem. Rev.* **70**, 471 (2001) reference.

4) On page 9, the authors estimate that the relative absorption intensities are accurate within a factor two. Can the authors explain their rationale for that statement?

This is based on calibration works by other groups (JCP 96, 9005 ; JCP 109, 10587; PCCP 10, 6621). These calibration studies are based on small molecules, as very few accurate experimental results are actually available, and these results suggest a substantially higher accuracy than a factor of two. For larger systems, and with explicit water molecules, our conservative estimate is a factor of two. We have modified the sentence to “Based on calibration studies, the relative absorption intensities are conservatively estimated to be accurate within a factor of two” and included the above mentioned references.

5) The 3D plots (plots in Fig. 2, Fig. 7a, plots in Fig. 9) would be way more readable and simpler to analyze by the reader if they were presented as 2D colormaps using a uniform color scheme like viridis (see 10.1038/s41467-020-19160-7).

We have added a Supplementary Information, where we present 2D colormaps of all the data presented as 3D plots in the manuscript. Reference to the supplementary figures are given in the pertinent figure captions in the manuscript. We prefer to keep the 3D plots in the manuscript, because the topology of 3D plots give a better dynamic range enabling the reader to see small changes in the absorption close to much stronger absorption dynamics. Compare for instance the carbon dioxide absorption dynamics shown in Fig. 7 and Fig. S12. The $v = 1 \rightarrow v = 2$ is clearly visible in the 3D plot but absent in the 2D contour. Moreover, the 3D plots give a quantitative measure of the transient absorption dynamics and thus enable the reader to estimate the important quantum yields derived from the data.

Reviewer #2 (Remarks to the Author):

This is a fine piece of work that studies aqueous pyruvate photochemistry dynamics at deep- as well as in near-UV. The authors made a convincing argument to justify the importance of this work under the context of the Earth's atmosphere and the origin of life. This work also completes our understanding of the photochemistry of pyruvate in gas-, cluster- and aqueous phase in the near-UV region, and extends to the deep-UV. The different photochemistry behaviors were reported at two excitations and quantum yields for certain products (e.g. CO₂ (aq)) were estimated. Both experiments and theoretical modellings were competently and carefully conducted. I only have a few minor quires for the authors: 1) on page 13, the quantum yield of photo-dissociation is given 0.5 +/-3%. It could be misunderstood as 50%. It would be better to express (0.5 +/-3)%; 2) page 15, the reference is 13 not 12 when referring to two water molecules blocking all dissociation channels.

We have changed the way we present the yields throughout the manuscript (page 12 and page 13), so they now read $Y = X \% \pm Z\%$.

We have changed reference 12 to reference 13 on page 15.

Reviewer #1 (Remarks to the Author):

The authors addressed most of my comments and questions. There are a few elements that would still require their attention.

Q1: My comment about the Tamm-Dancoff approximation was not related to the computational cost of the TDDFT calculations with this approximation but more to the fact that this approximation stabilizes the linear-response TDDFT equations when electronic states are close in energy (see the work by Casida: 10.1063/1.2978380 or 10.1063/1.2786997) – which is apparently one of the issues experienced by the authors and described in their reply to my comment 2.

The authors only partially addressed my comment on the microsolvated structure (with the two water molecules). Is the conformer obtained (and described) the lowest-energy conformer of pyruvate surrounded by two waters?

In the text added to the revised manuscript, the authors say: 'Excitation energies have been calculated within the TDDFT framework, where only the optical part of the dielectric constant is included.' It should be made clear that the second part of the sentence is related to the IEFPCM.

Q2: The authors mention that they did not perform a calibration study as the active space for CAS-type calculations would be large and trust the TDDFT results. I leave it to the authors' choice to decide whether they trust these calculations. Still, I must stress that such calibration calculations are (unfortunately) always required when exploring excited states to validate the quantum-chemical method used (particularly near conical intersections, as experienced by the authors). The authors mention 'avoided crossings', but I guess they want to say 'conical intersections' as the issue with TDDFT only occurs close to conical intersections between the ground state and the lowest excited state (see Casida's article above, 10.1063/1.2978380). References to 'avoided crossing' should be checked in the text – is 'conical intersection' meant?

REVIEWER COMMENTS

Reviewer #1 (Remarks to the Author):

The authors addressed most of my comments and questions. There are a few elements that would still require their attention.

Q1: My comment about the Tamm-Dancoff approximation was not related to the computational cost of the TDDFT calculations with this approximation but more to the fact that this approximation stabilizes the linear-response TDDFT equations when electronic states are close in energy (see the work by Casida: 10.1063/1.2978380 or 10.1063/1.2786997) – which is apparently one of the issues experienced by the authors and described in their reply to my comment 2.

The above two papers investigate the ring-opening of oxirane on the excited state surface, where the reference ground state becomes progressively multiconfigurational with increasing angle. The ring-opening reaction of oxirane thus leads to a state with biradical character, where both singlet and triplet coupling of the two radical sites are possible, and the decoupling by the TD approximation improves the excitation energies. The dissociation of CO₂ from pyruvate, in contrast, does not lead to radical products, as the dissociation limit corresponds to two closed shell molecules. For excitations from a closed shell reference state, the TD approximation normally gives results very similar to the full TDDFT, for a modest computational saving.

The G16 program package cannot employ the TD approximation. We have tested the difference in Gamess with the CAM3BLYP functional (wb97XD cannot be used for TDDFT in Gamess), and where the implicit solvent model is slightly different than in G16, although still using only the optical dielectric constant in the vertical excitation energy. At the relaxed excited state geometry for a C-C distance of 2.40 Å, the wb97XD TDDFT result is 1.84 eV, while the corresponding EOMCCSD value is 2.00 eV. With CAMB3LYP, the TDDFT result is 1.84 eV, which changes to 2.02 eV with the TD approximation. Based on this single point, the use of the TD approximation thus brings the DFT excitation energy in slightly better agreement with the EOMCCSD result. Nevertheless, since the present system corresponds to excitation from closed shell species at all distances, we do not believe that the use of the TD approximation would give significantly different results. The above calibration results suggest that the reaction barrier on the excited state surface would increase by ~0.15 eV, but this would make no difference in the conclusions.

The authors only partially addressed my comment on the microsolvated structure (with the two water molecules). Is the conformer obtained (and described) the lowest-energy conformer of pyruvate surrounded by two waters?

The isolated pyruvate has only a single conformation corresponding to the carboxyl and carbonyl groups being perpendicular to each other, but as mentioned in the manuscript, the barrier towards planarity is only a few kJ/mol. The optimized structure with two water molecules has an intermediate geometry with a torsional angle of 52 degrees. The optimized structure is shown below, where the dashed lines just are the particular visualization programs decision for where to draw hydrogen bonds. The two water molecules coordinate to pyruvate oxygens with OH distances of 1.76, 1.76, 2.46 and 2.89 Å. There is thus no ambiguity in the pyruvate conformation, and it is difficult to imagine a more favorable arrangement of two water molecules. We are thus fairly confident that this is the lowest energy structure for the microsolvated pyruvate. The picture below has been included in the SI.

In the text added to the revised manuscript, the authors say: 'Excitation energies have been calculated within the TDDFT framework, where only the optical part of the dielectric constant is included.' It should be made clear that the second part of the sentence is related to the IEFPCM.

Correct. The sentence has been changed to: "Excitation energies have been calculated within the TDDFT framework, where only the optical part of the dielectric constant in the IEFPCM model is included."

Q2: The authors mention that they did not perform a calibration study as the active space for CAS-type calculations would be large and trust the TDDFT results. I leave it to the authors' choice to decide whether they trust these calculations. Still, I must stress that such calibration calculations are (unfortunately) always required when exploring excited states to validate the quantum-chemical method used (particularly near conical intersections, as experienced by the authors).

Calculating bond dissociation curves normally require MCSCF based method, as most bonds dissociate homolytically into radical fragments. In the present case, however, the dissociation of CO₂ is into two closed shell fragments, and single determinant based methods should thus be valid. Whether TDDFT is a valid description of the excited state surface(s) is open to discussion. At the equilibrium ground state geometry for the isolated pyruvate, the calculated TDDFT excitation energies are 4.2 and 5.8 eV, and the corresponding EOMCCSD values with the aug-cc-pVDZ and aug-cc-pVTZ basis sets are 4.3 and 5.8 eV, and 4.3 and 5.9 eV, respectively. The DFT calculated ionization potential is 6.3 eV, while the CCSD(T)/aug-cc-pVTZ value is 6.3 eV. The DFT calculated triplet state is 3.6 eV, while the corresponding CCSD(T)/aug-cc-pVTZ value is 4.0 eV.

The corresponding excitation energies for the microsolvated system is 3.9 and 5.4 eV at the TDDFT level and 4.1 and 5.8 eV at the EOMCCSD/aug-cc-pVDZ level. The calculated DFT IP is 6.8 eV, while the CCSD(T)/aug-cc-pVDZ value is 6.6 eV. The triplet state is calculated at 3.4 eV with DFT and at 3.7 eV with CCSD(T)/aug-cc-pVDZ. These CC methods are considered as the currently best theoretical tools, and the good agreement with the DFT based method suggests that the (TD)DFT relative energies of excited singlet and triplet states, as well as the ionization potential, are valid at the equilibrium geometry.

The calculated IP for gas phase puruvate is 3.82 and 3.75 eV at the wB97XD level with the aug-pcseg-1 and -2 basis sets, 3.67 and 3.82 eV at the CCSD(T) level with the aug-cc-pVDZ and aug-cc-pVTZ basis sets, and these can be compared to the experimental result of 4.04 +/- 0.18 eV.

The calculation of conical intersections, or regions where two energy surfaces become close in energy, normally requires MCSCF based methods, and usually also including dynamical correlation by e.g. a PT2 or MR2 approach. Selecting an appropriate orbital space for the MCSCF part is often a trial-and-error exercise. Assuming that the pi-systems of the carboxyl and carbonyl moieties as well as the oxygen lone-pair orbitals must be included in the active space, and with at least one correlating orbital for each occupied, this suggests a CASSCF space of at least 18-electrons in 18-orbitals. This amounts to $\sim 2 \times 10^9$ determinants, which is out of reach for our computational resources, and would be a major task in any state-of-the-art computational facility.

The authors mention 'avoided crossings', but I guess they want to say 'conical intersections' as the issue with TDDFT only occurs close to conical intersections between the ground state and the lowest excited state (see Casida's article above, 10.1063/1.2978380). References to 'avoided crossing' should be checked in the text – is 'conical intersection' meant?

Since TDDFT cannot describe the region near a (true) conical intersection, the observation that the calculated TDDFT excitation energy approaches zero can only be taken as indirect evidence of either a conical intersection or an accidental near-degeneracy of the ground and first excited states. The TDDFT excitation energy at the excited state relaxed geometry with the distance 2.40 Å is 1.84 eV, while the EOMCCSD/aug-cc-pVDZ value is 2.00 eV. This suggests that the TDDFT approach is valid out to a distance of 2.40 Å. Increasing the distance to 2.45 Å, however, leads to a non-converging optimization, due to different geometry steps leading to different states being lowest in energy. We have modified the text to reflect this ambiguity. “The geometry corresponding to the relaxed excited singlet state at a distance of 2.4 Å has an excitation energy of only 1.8 eV. At longer distances the ground and excited states become so close in energy that attempts of geometry optimization fail, as predicted geometry steps lead to state switching. The employed TDDFT method cannot describe regions with near-degenerate states, and the observed behavior can thus only be taken as indirect evidence of either a conical intersection or an accidental near-degeneracy of the ground and excited states.”

We finally want to mention that the same computational procedure has been used by us in a number of related studies (refs 31-34), where similar calibration results have been obtained, and where the calculated results have aided in the interpretation of the experimental results.

Reviewer #1 (Remarks to the Author):

I thank the authors for their answers.

The Tamm-Dancoff approximation to the linear-response TDDFT equation is available in G16 and can be requested by using the keyword 'TDA' instead of 'TD'.

The added text 'The employed TDDFT method cannot describe regions with near-degenerate states' lacks accuracy as TDDFT can describe regions of near degeneracy between two excited states (see for example 10.1063/5.0176140). The authors may want to alter this sentence.

REVIEWER COMMENTS

Reviewer #1 (Remarks to the Author):

I thank the authors for their answers.

The Tamm-Dancoff approximation to the linear-response TDDFT equation is available in G16 and can be requested by using the keyword 'TDA' instead of 'TD'.

Correct, we missed that. The TDA result with the wb97XD functional is 2.02 eV and thus identical to the CAM-BLYP value calculated by Gamess, as given in the previous response to reviewer comments.

The added text 'The employed TDDFT method cannot describe regions with near-degenerate states' lacks accuracy as TDDFT can describe regions of near degeneracy between two excited states (see for example 10.1063/5.0176140. The authors may want to alter this sentence.

Correct. The main problem is with degeneracy between the ground and the first excited state, and the text has been modified accordingly: "TDDFT method cannot describe regions with near-degenerate states" has been changed to "TDDFT method cannot describe regions with near-degeneracy between the ground and first excited states"